# WATCH, TRY, LEARN: META-LEARNING FROM DEMONSTRATIONS AND REWARDS

**Allan Zhou**[*] **& Eric Jang**
Google Brain
{allanz,ejang}@google.com

**Daniel Kappler & Alex Herzog**
X
{kappler,alexherzog}@x.team

**Mohi Khansari, Paul Wohlart, Yunfei Bai & Mrinal Kalakrishnan**
X
{khansari,wohlhart,yunfeibai,kalakris}@x.team

**Sergey Levine**
Google Brain, UC Berkeley
slevine@google.com

**Chelsea Finn**
Google Brain, Stanford
chelseaf@google.com

## ABSTRACT

Imitation learning allows agents to learn complex behaviors from demonstrations. However, learning a complex vision-based task may require an impractical number of demonstrations. Meta-imitation learning is a promising approach towards enabling agents to learn a new task from one or a few demonstrations by leveraging experience from learning similar tasks. In the presence of task ambiguity or unobserved dynamics, demonstrations alone may not provide enough information; an agent must also try the task to successfully infer a policy. In this work, we propose a method that can learn to learn from both demonstrations and trial-and-error experience with sparse reward feedback. In comparison to meta-imitation, this approach enables the agent to effectively and efficiently improve itself autonomously beyond the demonstration data. In comparison to meta-reinforcement learning, we can scale to substantially broader distributions of tasks, as the demonstration reduces the burden of exploration. Our experiments show that our method significantly outperforms prior approaches on a set of challenging, vision-based control tasks.

## 1 INTRODUCTION

Imitation learning enables autonomous agents to learn complex behaviors from demonstrations, which are often easy and intuitive for users to provide. However, learning expressive neural network policies from imitation requires a large number of demonstrations, particularly when learning from high-dimensional inputs such as images. Meta-imitation learning has emerged as a promising approach for allowing an agent to leverage data from previous tasks in order to learn a new task from only a handful of demonstrations (Duan et al., 2017; Finn et al., 2017b; James et al., 2018). However, in many practical few-shot imitation settings, there is an identifiability problem: it may not be possible to precisely determine a policy from one or a few demonstrations, especially in a new situation. And even if a demonstration precisely communicates *what* the task entails, it might not precisely communicate *how* to accomplish it in new situations. For example, it may be difficult to discern from a single demonstration where to grasp an object when it is in a new position or how much force to apply in order to slide an object without knocking it over. It may be expensive to collect more demonstrations to resolve such ambiguities, and even when we can, it may not be obvious to a human demonstrator where the agent's difficulty is arising from. Alternatively, it is easy for the user to provide success-or-failure feedback, while exploratory interaction is useful for learning how to perform the task. To this end, our goal is to build an agent that can first infer a policy from

---

[*]Work done as a Google AI Resident.

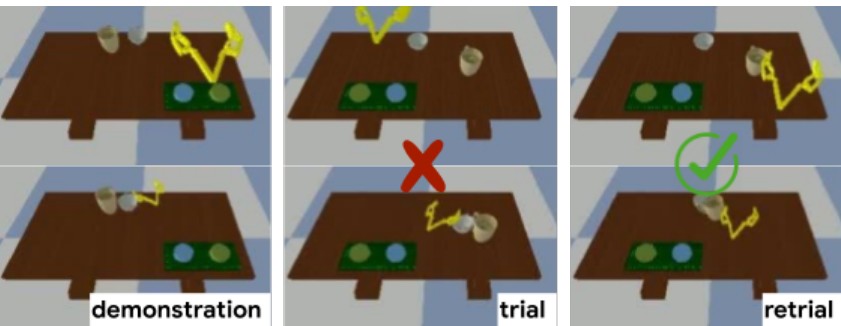

Figure 1: Each column displays the first and last frame of an episode top-to-bottom. After watching one demonstration (left), the scene is re-arranged. With one trial episode (middle), our method can learn to solve the task (right) by leveraging both the demo and trial-and-error experience.

one demonstration, then attempt the task using that policy while receiving binary user feedback, and finally use the feedback to improve its policy such that it can consistently solve the task.

This vision of learning new tasks from a few demonstrations and trials inherently requires some amount of prior knowledge or experience, which we can acquire through meta-learning across a range of previous tasks. To this end, we develop a new meta-learning algorithm that incorporates elements of imitation learning with trial-and-error reinforcement learning. In contrast to previous meta-imitation learning approaches that learn one-shot imitation learning procedures through imitation (Duan et al., 2017; Finn et al., 2017b), our approach enables the agent to improve at the test task through trial-and-error. Further, from the perspective of meta-RL algorithms that aim to learn efficient RL procedures (Duan et al., 2016; Wang et al., 2016; Finn et al., 2017a), our approach also has significant appeal: as we aim to scale meta-RL towards broader task distributions and learn increasingly general RL procedures, exploration and efficiency becomes exceedingly difficult. However, a demonstration can significantly narrow down the search space while also providing a practical means for a user to communicate the goal, enabling the agent to achieve few-shot learning of behavior. While the combination of demonstrations and reinforcement has been studied extensively in single task problems (Kober et al., 2013; Sun et al., 2018; Rajeswaran et al., 2018; Le et al., 2018), this combination is particularly important in meta-learning contexts where few-shot learning of new tasks is simply not possible without demonstrations. Further, we can even significantly improve upon prior methods that study this combination using meta-learning to more effectively integrate the information coming from both sources.

The primary contribution of this paper is a meta-learning algorithm that enables learning of new behaviors with a single demonstration and trial experience. After receiving a demonstration illustrating a new goal, the meta-trained agent can learn to accomplish that goal through a small amount of trial-and-error with only binary success-or-failure labels. We evaluate our algorithm and several prior methods on a challenging, vision-based control problem involving manipulation tasks from four distinct families of tasks: button-pressing, grasping, pushing, and pick and place. We find that our approach can effectively learn tasks with new, held-out objects using one demonstration and a single trial, while significantly outperforming meta-imitation learning, meta-reinforcement learning, and prior methods that combine demonstrations and reward feedback. To our knowledge, our experiments are the first to show that meta-learning can enable an agent to adapt to new tasks with binary reinforcement signals from raw pixel observations, which we show with a single meta-model for a variety of distinct manipulation tasks. We have published videos of our experimental results[1] and the experiment model code[2].

## 2 RELATED WORK

Learning to learn, or meta-learning, has a long-standing history in the machine learning literature (Thrun & Pratt, 1998; Schmidhuber, 1987; Bengio et al., 1992; Hochreiter et al., 2001). We particularly focus on meta-learning in the context of control. Our approach builds on and signifi-

---

[1] https://sites.google.com/view/watch-try-learn-project
[2] https://github.com/google-research/tensor2robot/tree/master/research/vrgripper

cantly improves upon meta-imitation learning (Duan et al., 2017; Finn et al., 2017b; James et al., 2018; Paine et al., 2018) and meta-reinforcement learning (Duan et al., 2016; Wang et al., 2016; Mishra et al., 2018; Rakelly et al., 2019), extending contextual meta-learning approaches. Unlike prior work in few-shot imitation learning (Duan et al., 2017; Finn et al., 2017b; Yu et al., 2018; James et al., 2018; Paine et al., 2018), our method enables the agent to additionally improve upon trial-and-error experience. In contrast to work in multi-task and meta-reinforcement learning (Duan et al., 2016; Wang et al., 2016; Finn et al., 2017a; Mishra et al., 2018; Houthooft et al., 2018; Sung et al., 2017; Nagabandi et al., 2019; Sæmundsson et al., 2018; Hausman et al., 2017), our approach learns to use one demonstration to address the meta-exploration problem (Gupta et al., 2018; Stadie et al., 2018). Our work also requires only one round of on-policy data collection, collecting only $1,500$ trials for the vision-based manipulation tasks, while nearly all prior meta-learning works require thousands of iterations of on-policy data collection, amounting to hundreds of thousands of trials (Duan et al., 2016; Wang et al., 2016; Finn et al., 2017a; Mishra et al., 2018).

Combining demonstrations and trial-and-error experience has long been explored in the machine learning and robotics literature (Kober et al., 2013). This ranges from simple techniques such as demonstration-based pre-training and initialization (Peters & Schaal, 2006; Kober & Peters, 2009; Kormushev et al., 2010; Kober et al., 2013; Silver et al., 2016) to more complex methods that incorporate both demonstration data and reward information in the loop of training (Taylor et al., 2011; Brys et al., 2015; Subramanian et al., 2016; Hester et al., 2018; Sun et al., 2018; Rajeswaran et al., 2018; Nair et al., 2018; Le et al., 2018). The key contribution of this paper is an algorithm that can *learn* how to learn from both demonstrations and rewards. This is quite different from RL algorithms that incorporate demonstrations: learning from scratch with demonstrations and RL involves a slow, iterative learning process, while fast adaptation with a meta-trained policy involves extracting inherently distinct pieces of information from the demonstration and the trials. The demonstration provides information about *what* to do, while the small number of RL trials can disambiguate the task and how it s refinement. As a result, we get a procedure that significantly exceeds the efficiency of prior approaches, requiring only one demonstration and one trial to adapt to a new test task, even from pixel observations, by leveraging previous data. In comparison, single-task methods for learning from demonstrations and rewards typically require hundreds or thousands of trials to learn tasks of comparable difficulty (Rajeswaran et al., 2018; Nair et al., 2018).

## 3 META-LEARNING FROM DEMONSTRATIONS AND REWARDS

We first introduce some meta-learning preliminaries, then formalize our particular problem statement before finally describing our approach and its implementation.

### 3.1 PRELIMINARIES

Meta-learning, or learning to learn, aims to learn new *tasks* from very little data. To achieve this a meta-learning algorithm can first *meta-train* on a set of tasks $\{\mathcal{T}_i\}$, called the meta-train tasks. We then evaluate how quickly the meta-learner can learn an unseen *meta-test* task $\mathcal{T}_j$. We typically assume that the meta-train and meta-test tasks are drawn from some unknown task distribution $p(\mathcal{T})$ (Finn, 2018). Through meta-training, a meta-learner can learn some common structure between the tasks in $p(\mathcal{T})$ which it can use to more quickly learn a new meta-test task.

We define a task $\mathcal{T}_i$ as a finite-horizon Markov decision process (MDP), $\{\mathcal{S}, \mathcal{A}, r_i, P_i, H\}$, with continuous state space $\mathcal{S}$, continuous action space $\mathcal{A}$, reward function $r_i : \mathcal{S} \times \mathcal{A} \rightarrow \mathbb{R}$, unknown dynamics $P_i(s_{t+1}|s_t, a_t)$, and horizon $H$. In our manipulation experiments we will restrict the tasks to sparse binary rewards $r_i : \mathcal{S} \rightarrow \{0, 1\}$. The state space $\mathcal{S}$, reward $r_i$, and dynamics $P_i$ may vary across tasks.

### 3.2 PROBLEM STATEMENT

Our goal is to meta-train an agent such that can quickly learn a new test task $\mathcal{T}_j$ in two phases:

**Phase I**: The agent observes and learns from $k = 1, ..., K$ task demonstrations $\mathcal{D}_j^* := \{\mathbf{d}_{j,k}\}$. It can then attempt the task in $\ell = 1, \cdots, L$ *trial episodes* $\{\boldsymbol{\tau}_{j,\ell}\}$, for which it receives reward labels.

**Phase II**: The agent learns from those trial episodes *and* the original demos to succeed at $\mathcal{T}_j$.

A demonstration is a trajectory $\mathbf{d} = \{(s_t, a_t)\}$ of $T$ states-action tuples that succeeds at the task, while a trial episode is a trajectory $\boldsymbol{\tau} = \{(s_t, a_t, r_i(s_t, a_t))\}$ that also contains reward information.

### 3.3 LEARNING TO IMITATE AND TRY AGAIN

With the aforementioned problem statement in mind, we aim to develop a method that can learn to learn from both demonstration and trial-and-error experience. We wish to (1) meta-learn a Phase I policy that is suitable for gathering information about a task given demonstrations, and (2) meta-learn a Phase II policy which learns from both demonstrations and trials produced by the Phase I policy. Like prior few shot meta-learning works (Finn et al., 2017b; Duan et al., 2017), we hope to achieve few shot success on a task by explicitly meta-training our Phase I and Phase II policies to learn from very little demonstration and trial data.

We write the Phase I policy as $\pi_\theta^{\mathrm{I}}(a|s, \{\mathbf{d}_{i,k}\})$, where $\theta$ represents all the learnable parameters. $\pi^{\mathrm{I}}$ conditions on the task demonstrations $\{\mathbf{d}_{i,k}\}$ which helps it infer what the unknown task is before attempting the trials. We condition the Phase II policy on both demonstration and trial data and write it as $\pi_\phi^{\mathrm{II}}(a|s, \{\mathbf{d}_{i,k}\}, \{\boldsymbol{\tau}_{i,\ell}\})$, with parameters $\phi$. One simple yet naive approach would be to use a single model across both phases: e.g., using a single MAML (Finn et al., 2017a) policy that sees the demonstration, executes trial(s) in the environment and then adapts its own weights from them. However, a key challenge in meta-training such a single model is that updates based on Phase II behavior (after the trials) will also change Phase I behavior (during the trials). Thus each meta-training update changes the distribution of trial trajectories $\boldsymbol{\tau}_{i,\ell} \sim \pi_\theta^{\mathrm{I}}(a|s, \{\mathbf{d}_{i,k}\})$ that the Phase II policy learns from. Prior meta-reinforcement learning work (Duan et al., 2016; Finn et al., 2017a) have addressed the issue of a changing trial distribution by re-collecting *on-policy* trial trajectories from the environment after every gradient step during meta-training, but this can be difficult in real-world problem settings with broad task distributions, where it is impractical to collect large amounts of on-policy experience.

Instead, we represent and train $\pi_\theta^{\mathrm{I}}, \pi_\phi^{\mathrm{II}}$ separately, decoupling their optimization. In particular, we train $\pi_\theta^{\mathrm{I}}$ first, freeze its weights, and collect trial data $\{\boldsymbol{\tau}_{i,\ell}\}$ from the environment for each meta-training task $\mathcal{T}_i$. We then train $\pi_\phi^{\mathrm{II}}$ using our collected trial data. Crucially, $\theta$ and $\phi$ are separate so training $\pi_\phi^{\mathrm{II}}$ will not change $\pi_\theta^{\mathrm{I}}$ behavior, and the distribution of trial data $\{\boldsymbol{\tau}_{i,l}\}$ for each task remains stationary. How do we train each of these policies with off-policy demonstration and trial data? $\pi^{\mathrm{I}}$ must be trained in a way that will provide useful exploration for inferring the task. One simple and effective strategy for exploration is posterior or Thompson sampling (Russo et al., 2018; Rakelly et al., 2019), i.e. greedily act according to the policy's current belief of the task. To this end, we train $\pi^{\mathrm{I}}$ using a meta-imitation learning setup, where for each task $\mathcal{T}_i$ we assume access to a set of demonstrations $\mathcal{D}_i^*$. $\pi^{\mathrm{I}}$ conditions on $K$ demonstrations $\{\mathbf{d}_{i,k}\} \subset \mathcal{D}_i^*$ and aims to

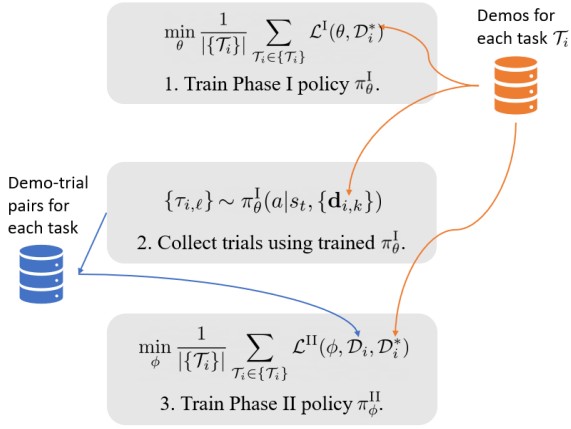

Figure 2: **Meta-training Overview**: First, we meta-train $\pi_\theta^{\mathrm{I}}$ according to Eq. 2. Next, we collect $L$ trial trajectories per meta-training task $\mathcal{T}_i$ in the environment using our trained $\pi_\theta^{\mathrm{I}}$. We denote the trial $\{\boldsymbol{\tau}_{i,\ell}\} \sim \pi_\theta^{\mathrm{I}}(a|s, \{\mathbf{d}_{i,k}\})$, and store the resulting demo-trial pairs $\{(\{\mathbf{d}_{i,k}\}, \{\boldsymbol{\tau}_{i,j}\})\}$ in a new dataset (blue). Finally, we meta-train $\pi_\phi^{\mathrm{II}}$ according to Eq. 4.

maximize the likelihood of the actions under another demonstration of the same task $\mathbf{d}_i^{\mathrm{test}} \in \mathcal{D}_i^*$ (the test demo is distinct from the conditioning demos). This gives the following Phase I loss for $\mathcal{T}_i$:

$$\mathcal{L}^{\mathrm{I}}(\theta, \mathcal{D}_i^*) = \mathbb{E}_{\{\mathbf{d}_{i,k}\} \sim \mathcal{D}_i^*} \mathbb{E}_{\mathbf{d}_i^{\mathrm{test}} \sim \mathcal{D}_i^* \setminus \{\mathbf{d}_{i,k}\}} \mathbb{E}_{(s_t, a_t) \sim \mathbf{d}_i^{\mathrm{test}}} \left[ -\log \pi_\theta^{\mathrm{I}}(a_t|s_t, \{\mathbf{d}_{i,k}\}) \right] \quad (1)$$

Then we can meta-train $\pi_\theta^{\mathrm{I}}$ by minimizing Eq. 1 across the set of meta-train tasks $\{\mathcal{T}_i\}$:

$$\min_\theta \frac{1}{|\{\mathcal{T}_i\}|} \sum_{\mathcal{T}_i \in \{\mathcal{T}_i\}} \mathcal{L}^{\mathrm{I}}(\theta, \mathcal{D}_i^*) \quad (2)$$

We train $\pi^{\mathrm{II}}$ in a similar fashion, but additionally condition on $L$ trial trajectories $\{\boldsymbol{\tau}_{i,\ell}\}$, which are the result of executing $\pi^{\mathrm{I}}$ in the environment. Suppose for any task $\mathcal{T}_i$, we have a set of demo-trial

**Algorithm 1** Watch-Try-Learn: Meta-training

1: **Input:** Training tasks $\{\mathcal{T}_i\}$
2: **Input:** Demo data $\mathcal{D}_i^* = \{\mathbf{d}_i\}$ per task $\mathcal{T}_i$
3: **Input:** Number of training steps $N$
4: Randomly initialize $\theta, \phi$
5: **for** step $= 1, \cdots, N$ **do**
6:      Sample meta-training task $\mathcal{T}_i$
7:      Update $\theta$ with $\nabla_\theta \mathcal{L}^{\mathrm{I}}(\theta, \mathcal{D}_i^*)$ (see Eq. 1)
8: **end for**
9: **for** $\mathcal{T}_i \in \{\mathcal{T}_i\}$ **do**
10:      Initialize empty $\mathcal{D}_i$ for demo-trial pairs.
11:      **while** not done **do**
12:          Sample $K$ demonstrations $\{\mathbf{d}_{i,k}\} \sim \mathcal{D}_i^*$
13:          Collect $L$ trials $\{\boldsymbol{\tau}_{i,l}\} \sim \pi_\theta^T(a|s, \{\mathbf{d}_{i,k}\})$
14:          Update $\mathcal{D}_i \leftarrow \mathcal{D}_i \cup \{(\{\mathbf{d}_{i,k}\}, \{\boldsymbol{\tau}_{i,l}\})\}$
15:      **end while**
16: **end for**
17: **for** step $= 1, \cdots, N$ **do**
18:      Sample meta-training task $\mathcal{T}_i$
19:      Update $\phi$ with $\nabla_\phi \mathcal{L}^{\mathrm{II}}(\phi, \mathcal{D}_i, \mathcal{D}_i^*)$ (see Eq. 3)
20: **end for**
21: **return** $\theta, \phi$

**Algorithm 2** Watch-Try-Learn: Meta-testing

1: **Input:** Test tasks $\{\mathcal{T}_j\}$
2: **Input:** Demo data $D_j^*$ for task $\mathcal{T}_j$
3: **for** $\mathcal{T}_j \in \{\mathcal{T}_j\}$ **do**
4:      Sample $K$ demonstrations $\{\mathbf{d}_{j,k}\} \sim \mathcal{D}_j^*$
5:      Collect $L$ trials $\{\boldsymbol{\tau}_{j,l}\}$ with policy $\pi_\theta^{\mathrm{I}}(a|s, \{\mathbf{d}_{j,k}\})$
6:      Perform task with re-trial policy $\pi_\phi^{\mathrm{II}}(a|s, \{\mathbf{d}_{j,k}\}, \{\boldsymbol{\tau}_{j,l}\})$
7: **end for**

pairs $\mathcal{D}_i = \{(\{\mathbf{d}_{i,k}\}, \{\boldsymbol{\tau}_{i,j}\})\}$. Then the Phase II objective for $\mathcal{T}_i$ is:

$$\mathcal{L}^{\mathrm{II}}(\phi, \mathcal{D}_i, \mathcal{D}_i^*) = \mathbb{E}_{(\{\mathbf{d}_{i,k}\}, \{\boldsymbol{\tau}_{i,\ell}\}) \sim \mathcal{D}_i} \mathbb{E}_{\mathbf{d}_i^{\mathrm{test}} \sim \mathcal{D}_i^* \setminus \{\mathbf{d}_{i,k}\}} \mathbb{E}_{(s_t, a_t) \sim \mathbf{d}_i^{\mathrm{test}}} \left[ -\log \pi_\phi^{\mathrm{II}}(a_t|s_t, \{\mathbf{d}_{i,k}\}, \{\boldsymbol{\tau}_{i,\ell}\}) \right] \tag{3}$$

Eq. 3 encourages $\pi^{\mathrm{II}}$ to use and improve upon the trial experience $\{\boldsymbol{\tau}_{i,\ell}\}$, which includes reward information. If a trial has high reward, then $\pi^{\mathrm{I}}$ likely inferred the task correctly from demonstration alone, and $\pi^{\mathrm{II}}$ should *reinforce that high reward trial behavior*. If a trial has low reward, then $\pi^{\mathrm{I}}$ probably inferred the task incorrectly and $\pi^{\mathrm{II}}$ should *avoid that low reward trial behavior*. Hence even failed trials can help $\pi^{\mathrm{II}}$ correctly infer the task by showing it what *not* to do. Note that since Eq. 3 evaluates log likelihood at states and actions from the *held out* demonstration $\mathbf{d}_i^{\mathrm{test}}$, $\pi^{\mathrm{II}}$ cannot simply copy behavior from high reward trials in $\{\boldsymbol{\tau}_{i,\ell}\}$ but instead must generalize from the trials to a new instance of the same task. We meta-train $\pi_\phi^{\mathrm{II}}$ by minimizing Eq. 3 across the meta-train tasks:

$$\min_\phi \frac{1}{|\{\mathcal{T}_i\}|} \sum_{\mathcal{T}_i \in \{\mathcal{T}_i\}} \mathcal{L}^{\mathrm{II}}(\phi, \mathcal{D}_i, \mathcal{D}_i^*) \tag{4}$$

We refer to our approach as Watch-Try-Learn (WTL), and describe our meta-training and meta-test procedures in detail in Alg. 1 and Alg. 2, respectively. We also illustrate the meta-training flow in Fig. 2. In practice we meta-train $\pi_\theta^{\mathrm{I}}$ and $\pi_\theta^{\mathrm{II}}$ by solving Eqs. 2 and 4 using stochastic gradient descent or some variant. We iteratively sample (minibatches of) tasks $\mathcal{T}_i$ to compute gradient updates on $\theta$ or $\phi$, respectively. At meta-test time, for any test task $\mathcal{T}_j$ we receive demonstrations $\{\mathbf{d}_{j,k}\}$ and obtain the demo-conditioned Phase I policy $\pi_\theta^{\mathrm{I}}(a|s, \{\mathbf{d}_{j,k}\})$. We collect trial(s) $\{\boldsymbol{\tau}_{j,l}\}$ using $\pi_\theta^{\mathrm{I}}$ in the environment. Then we obtain the Phase II policy $\pi_\phi^{\mathrm{II}}(a|s, \{\mathbf{d}_{j,k}\}, \{\boldsymbol{\tau}_{j,l}\})$. Finally, we execute the $\pi_\phi^{\mathrm{II}}$ in the environment to solve the task.

### 3.4 WATCH-TRY-LEARN IMPLEMENTATION

WTL and Alg. 1 allow for general representations of the Phase I policy $\pi_\theta^{\mathrm{I}}(a|s, \{\mathbf{d}_{i,k}\})$ so long as it conditions on the task demonstrations $\{\mathbf{d}_{i,k}\}$. Similarly, the Phase II policy $\pi^{\mathrm{II}}(a|s, \{\mathbf{d}_{i,k}\}, \{\boldsymbol{\tau}_{i,\ell}\})$ must condition on both $\{\mathbf{d}_{i,k}\}$ and the trials $\{\boldsymbol{\tau}_{i,\ell}\}$. Hence a variety of adaptation mechanisms could be used.

We choose to implement this conditioning in each policy by embedding the demonstration data and (for Phase II) the trial data into context vectors using neural networks. Figure 3 illustrates the $\pi_\phi^{\mathrm{II}}$ architecture assuming a single demonstration and trial. The embedding network first applies a

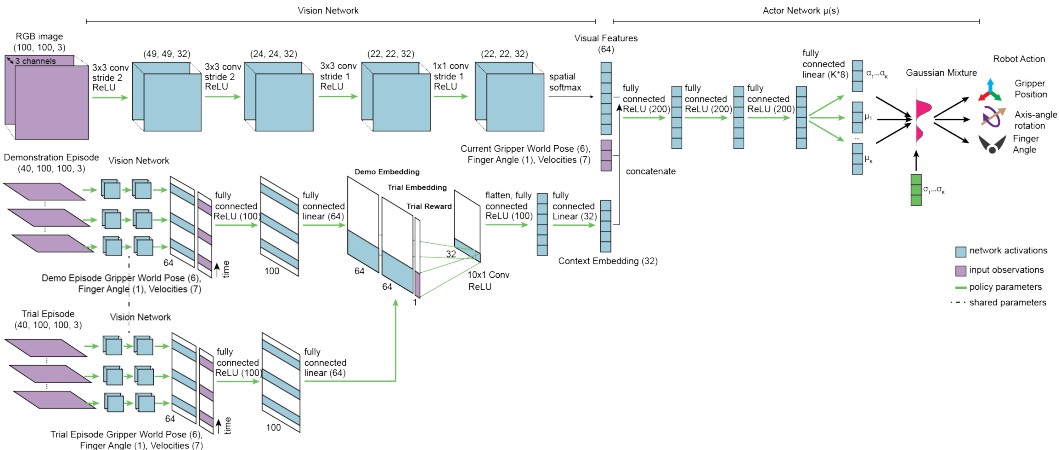

Figure 3: Our vision-based Phase II architecture: **Upper left**: we pass the RGB observation for each timestep through a 4-layer CNN with ReLU activations and layer normalization, followed by a spatial softmax layer that extracts 2D keypoints (Levine et al., 2016). We flatten the output keypoints and concatenate them with the current gripper pose, gripper velocity, and context embedding. **Upper Right**: We pass the resulting vector through the actor network, which predicts the parameters of a Gaussian mixture over the commanded end-effector position, axis-angle orientation, and finger angle. **Lower left**: To produce the context embedding, the embedding network applies a vision network to 40 ordered observations sampled randomly from the demo and trial trajectories. We concatenate the demo and trial outputs with the trial episode rewards along the embedding feature dimension, then apply a 10x1 convolution across the time dimension, flatten, and apply a MLP to produce the final context embedding. The Phase I policy architecture (Appendix Fig. 8) is the same as shown here, but omits the concatenation of trial embeddings and trial rewards.

vision network to each demonstration and trial trajectory image $s_t$, producing demo and trial feature matrices (respectively) where each row corresponds to one timestep's features. We concatenate the demo and trial feature matrices together on the feature (horizontal) dimension, along with the trial episode rewards to produce a single matrix combining demo and trial observation information and trial reward information. We then apply a 1-D convolution along the time (vertical) dimension of this matrix and flatten to obtain a single vector that integrates and aggregates information across time. Finally we apply a small MLP to produce the context embedding vector which contains information about both the demos $\{\mathbf{d}_{i,k}\}$ and the trials $\{\boldsymbol{\tau}_{i,\ell}\}$. The Phase I policy $\pi_\theta^{\mathrm{I}}$, illustrated in Appendix Figure 8, produces a similar context embedding but only uses demonstration data. This architectural design resembles prior contextual meta-learning works (Duan et al., 2016; Mishra et al., 2018; Duan et al., 2017; Rakelly et al., 2019), which have previously considered how to meta-learn efficiently from one modality of data (trials or demonstrations), but not how to integrate multiple sources, including off-policy trial data.

Since each policy is additionally conditioned on the current state $s$, we concatenate the context embedding with the current state features before feeding both as input to an actor network, which produces a Gaussian mixture distribution (Bishop, 1994) over actions. The current state features include visual features produced by another vision network, distinct from the one used to produce the context embedding. Both vision networks use a fairly standard convolutional neural network architecture with a final spatial softmax layer that extracts keypoints (Levine et al., 2016). The entire $\pi_\phi^{\mathrm{II}}$ architecture, illustrated in Figure 3, is trainable end-to-end using backpropagation on Eq. 4, where the parameters $\phi$ represent the collective weights of all layers. Similarly, the entire $\pi_\theta^{\mathrm{I}}$ architecture depicted in Appendix Figure 8 is trained by backpropagation on Eq. 2 where $\theta$ represents the set of weights across all layers. Note that since each neural network architecture is fixed and shared across tasks, we expect the input state dimensions to be the same across tasks, though the content (for example, the objects in the scene) may vary.

## 4 EXPERIMENTS

In our experiments, we aim to evaluate our method on challenging few-shot learning domains that span multiple task families, where the agent must use both demonstrations and trial-and-error to effectively infer a policy for the task. Prior meta-imitation benchmarks (Duan et al., 2017; Finn

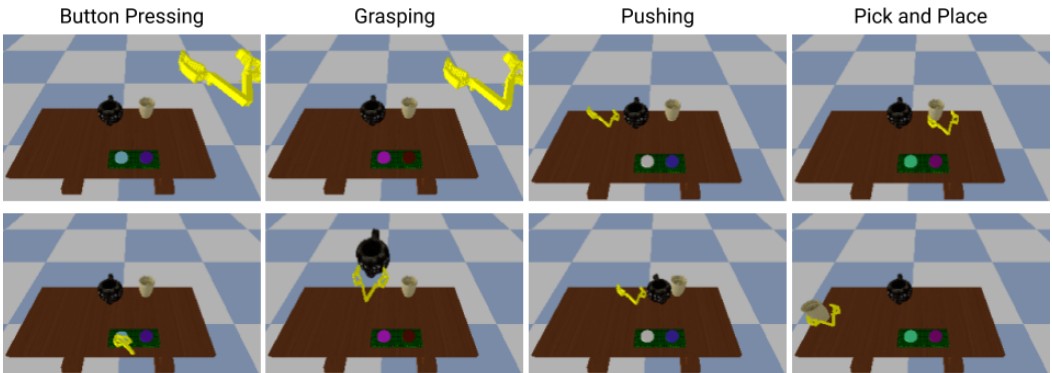

Figure 5: Illustration of example episodes in four distinct task families: button pressing, grasping, sliding, and pick-and-place. Each column shows the first and last frames of an episode top-to-bottom. We meta-train each model on hundreds of tasks from each of these task families. For each task within a task family, we use a unique pair of kitchenware objects sampled from a set of nearly one hundred different objects.

et al., 2017b) generally contain only a few tasks, and these tasks can be easily disambiguated given a single demonstration. Meanwhile, prior meta-reinforcement learning benchmarks (Duan et al., 2016; Finn et al., 2017a) tend to contain fairly similar tasks that a meta-learner can solve with little exploration and no demonstration at all. Motivated by these shortcomings, we design two new problems where a meta-learner can leverage a combination of demonstration and trial experience: a toy reaching problem and a challenging multitask gripper control problem, described below. We evaluate how the following methods perform in those environments:

**BC**: A behavior cloning method that does not condition on either demonstration or trial-and-error experience, trained across all meta-training data. We train BC policies using maximum log-likelihood with expert demonstration actions.

**MIL** (Finn et al., 2017b; James et al., 2018): A meta-imitation learning method that conditions on demonstration data, but does not leverage trial-and-error experience. We train MIL policies to minimize Eq. 1 similar to the WTL Phase I policy, but MIL methods lack a Phase II step. To perform a controlled comparison, we use the same architecture for both MIL and WTL.

**WTL**: Our Watch-Try-Learn method, which conditions on demonstration and trial experience. In all experiments, the agent receives $K = 1$ demonstration and can take $L = 1$ trial.

**BC + SAC**: In the gripper environment we study how much trial-and-error experience soft actor critic (SAC) (Haarnoja et al., 2018), a state of the art reinforcement learning algorithm, would require to solve a single task. While WTL meta-learns a single model that needs just one trial episode per meta-test task, in "BC + SAC" we fine-tune a separate RL agent for each meta-test task and analyze how much trial experience it needs to match WTL's single trial performance. We pre-train a policy similar to **BC**, then fine-tune for each meta-test task using SAC.

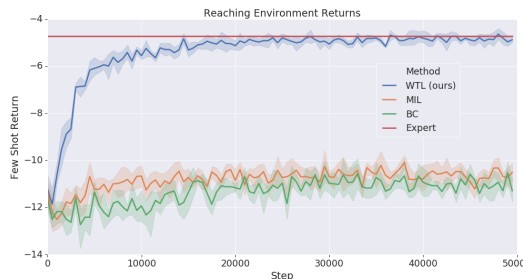

Figure 4: Average return of each method on held out meta-test tasks in the reaching environment, after one demonstration and one trial. Our Watch-Try-Learn (WTL) method is quickly able to learn to imitate the demonstrator. Each line shows the average over 5 separate training runs with identical hyperparameters, evaluated on 50 randomly sampled meta-test tasks. Shaded regions indicate 95% confidence intervals.

## 4.1 REACHING ENVIRONMENT EXPERIMENTS

To first verify that our method can actually leverage demonstration and trial experience in a simplified problem domain, we begin with toy planar reaching tasks inspired by Finn et al. (2017b) and illustrated in Appendix Fig 7. A demonstrator shows which of two objects to reach towards, but the agent's dynamics are randomized per task and may not match the demonstrator's. This simulates a domain adaptive setting such as a robot imitating a video of a human. Since the demonstrations $\{\mathbf{d}_{i,k}\}$ do not help identify the unknown

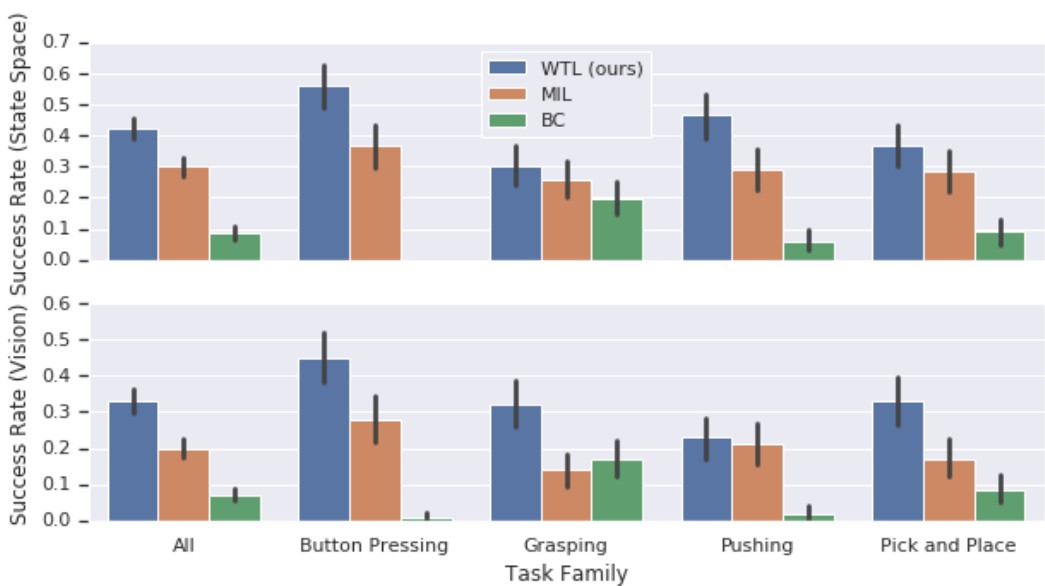

Figure 6: The average success rate of different methods in the gripper control environment, for both state space (non-vision) and vision based policies. The leftmost column displays aggregate results across all task families. Our Watch-Try-Learn (WTL) method significantly outperforms the meta-imitation (MIL) baseline, which in turn outperforms the behavior cloning (BC) baseline. We conducted 5 training runs of each method with identical hyperparameters and evaluated each run on 40 held out meta-test tasks. Error bars indicate 95% confidence intervals.

task dynamics, the agent must use trial episodes to successfully reach the target object. During meta-training, WTL meta-learns from demonstrations with the correct dynamics ($\mathbf{d}_i^{\text{test}}$ in Eq. 3) how to adapt to unknown dynamics in new tasks. To obtain expert demonstration data, we first train a reinforcement learning agent using normalized advantage functions (Gu et al., 2016), where the agent receives oracle observations that show only the true target. With our trained expert demonstration agent, we collect 2 demonstrations per task for 10000 meta-training tasks and 1000 meta-test tasks. For these toy experiments we use simplified versions of the architecture in Fig. 3 as described in Appendix C.

The results in Fig 4 show WTL is able to quickly learn to imitate the expert, while methods that do not leverage trial information struggle due to the uncertainty in the task dynamics.

## 4.2 GRIPPER ENVIRONMENT EXPERIMENTS

The gripper environment is a realistic 3-D simulation as shown in Figure 5. Gripper tasks fall into four broad task families: button pressing, grasping, pushing, and pick and place. Within a task family, each task involves a different pair of kitchenware objects sampled from a set of nearly one hundred. Unlike prior meta-imitation learning benchmarks, tasks of the same family that are qualitatively similar still have subtle but consequential differences. Some pushing tasks might require the agent to always push the left object towards the right object, while others might require the agent to always push, for example, the cup towards the teapot. The agent controls a free floating gripper with 7-D action space. The vision based policies receive image observations and gripper state, while state-space policies receive a vector of the gripper state and poses for all non-fixed objects in the scene. Gripper environment episodes have a maximum length of 5 seconds and contain sparse binary rewards. In an HTC Vive virtual reality setup, a human demonstrator recorded 1536 demonstrations for 768 distinct tasks involving 96 distinct sets of kitchenware objects. We held out 40 tasks corresponding to 5 sets of kitchenware objects for our meta-validation dataset, which we used for hyperparameter selection. Similarly, we selected and held out 5 object sets of 40 tasks for our meta-test dataset, which we used for final evaluations. Refer to Appendix A for a more detailed description of the scene setup, task families, and reward functions.

We trained and evaluated MIL, BC, and WTL policies with both state-space observations and vision observations. Appendix C describes hyperparameter selection using the meta-validation tasks and Appendix C.3 analyzes the sample and time complexity of WTL. The MIL policy uses an identical

architecture and objective to the WTL trial policy, while the BC policy architecture is the same as the WTL trial policy without the any embedding components. For vision based models, we crop and resize image observations from $300 \times 220$ to $100 \times 100$ before providing them as input.

We show the meta-test task success rates in Fig. 6. Overall, in both state space and vision domains, we find that WTL outperforms MIL and BC by a substantial margin, indicating that it can effectively leverage information from the trial and integrate it with that of the demonstration in order to achieve greater performance.

Finally, for the BC + SAC comparison we pre-trained an actor with behavior cloning and fine-tuned 4 RL agents per task with identical hyperparameters using the TFAgents (Guadarrama et al., 2018) SAC implementation. Table 1 shows that BC + SAC fine-tuning typically requires thousands of trial episodes *per task* to reach the same performance our meta-trained WTL method achieves after one demonstration and a single trial episode. Appendix C.4 shows the BC + SAC training curves averaged across the different meta-test tasks.

## 5 Discussion and Future Work

We proposed a meta-learning algorithm that allows an agent to quickly learn new behavior from a single demonstration followed by trial experience and associated (possibly sparse) rewards. The demonstration allows the agent to infer the type of task to be performed, and the trials enable it to improve its performance by resolving ambiguities in new test time situations. We presented experimental results where the agent is meta-trained on a broad distribution of tasks, after which it is able to quickly learn tasks with new held-out objects from just one demonstration and a trial. We showed that our approach outperforms prior meta-imitation approaches in challenging experimental domains.

As illustrated in our qualitative failure analysis (Appendix D), one area of future improvement involves improving the informativeness of even

| Method | Success Rate |
|---|---|
| BC | $.09 \pm .01$ |
| MIL | $.30 \pm .02$ |
| WTL, 1 trial (ours) | $.42 \pm .02$ |
| RL fine-tuning with SAC | |
| BC + SAC, 1500 trials | $.11 \pm .07$ |
| BC + SAC, 2000 trials | $.29 \pm .10$ |
| BC + SAC, 2500 trials | $.39 \pm .11$ |

Table 1: Average success rates across meta-test tasks using state space observations. For BC + SAC we pre-train with behavior cloning and use RL to fine-tune a separate agent on each meta-test task. The table shows BC + SAC performance after 1500, 2000, and 2500 trials *per task*.

failed trial trajectories generated by $\pi^{I}$. In future work, we hope to explore alternatives to posterior sampling that produce more informative trials while maintaining sample and computational efficiency. We also plan to explore ways of extending our approach to be meta-trained on a much broader range of tasks, testing the performance of the agent on completely new held-out tasks rather than on held-out objects.

The Watch-Try-Learn (WTL) approach enables a natural way for non-expert users to train agents to perform new tasks: by demonstrating the task and then observing and critiquing the performance of the agent on the task if it initially fails. WTL achieves this through a unique combination of demonstrations and trials in the inner loop of a meta-learning system, where the demonstration guides the exploration process for subsequent trials, and the use of trials allows the agent to learn new task objectives which may not have been seen during meta-training. We hope that this work paves the way towards more practical and general algorithms for meta-learning behavior.

## 6 Acknowledgements

We would like to thank Luke Metz and Archit Sharma for reviewing an earlier draft of this paper, and Alex Irpan for valuable discussions. We would also like to thank Murtaza Dalal for finding and correcting an error in the "BC+SAC" experimental results from an earlier version of this paper.

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

# A  GRIPPER ENVIRONMENT DETAILS

## A.1  SCENE SETUP

We created the gripper environment using the Bullet physics engine (Coumans & Bai, 2016). The agent controls a floating gripper that can move and rotate freely in space (6 DOFs), and 2 symmetric fingers that open and close together (1 DOF). For vision-based policies, the environment provides $300 \times 220$ RGB image observations and the 7-D gripper position vector at each timestep. For state-space policies it provides the 7-D gripper state and the 6-D poses (translation + rotation) of all non-fixed objects in the scene. The virtual scene is set up with the same viewpoint and table as in Figure 5 with random initial positions of the gripper, as well as two random kitchenware objects and a button panel that are placed on the table. The button panel contains two press-able buttons of different colors. One of the two kitchenware objects is placed onto a random location on the left half of the table, and the other kitchenware object is placed onto a random location on the right half of the table. After the demonstration episode of each task, the kitchenware objects and button panel are repositioned: a kitchenware object on the left half of the table moves to a random location on the right half of the table, and vice versa. Similarly, the two colored buttons in the button panel swap positions. Swapping the lateral positions of objects and buttons after the demonstration is crucial because otherwise, for example, the difference between the two types of pushing tasks would be meaningless.

## A.2  TASK FAMILIES

Our gripper environment has four broad task families: button pressing, grasping, sliding, and pick and place. But tasks in each family may come in one of two types:

- Button pressing 1: the demonstrator presses the left (respectively, right) button, and the agent must press the left (respectively, right) button.
- Button pressing 2: the demonstrator presses one of the two buttons, and the agent must press the button of the same color.
- Grasping 1: the demonstrator grasps and lifts the left (respectively, right) object. The agent must grasp and lift the left (respectively, right) object.
- Grasping 2: the demonstrator grasps and lifts one of the two objects. The agent must grasp and lift that same object.
- Sliding 1: the demonstrator pushes the left object into the right object (respectively, the right object into the left object). The agent must push the left object into the right object (respectively, the right object into the left object).
- Sliding 2: the demonstrator pushes one object (A) into the other (B). The agent must push object A into object B.
- Pick and Place 1: the demonstrator picks up one of the objects and places it on the near (respectively, far) edge of the table. The agent must pick up the same object and place it on the near (respectively, far) edge of the table.
- Pick and Place 2: the demonstrator picks up one of the objects and places it on the left (respectively, right) edge of the table. The agent must pick up the same object and place it on the left (respectively, right) edge of the table.

## A.3  EPISODES AND REWARDS

A gripper environment episode has a maximum length of 50 timesteps, or 5 seconds real time. At each timestep the environment returns a reward of $+1$ if the agent successfully completes the task, or 0 otherwise. The episode ends immediately upon success, so only the terminal timestep can have nonzero reward. Computing success (and hence the reward) depends on the current task's family. For each task family, the environment determines success at the current timestep if:

- Button pressing: the gripper is in contact with the correct button.
- Grasping: the correct object's vertical position is a certain threshold above the table.
- Sliding: the two objects are in contact, and object A has moved more than object B.
- Pick and Place: the correct object touches the correct edge of the table, and has cleared some vertical threshold at some prior point in the episode.

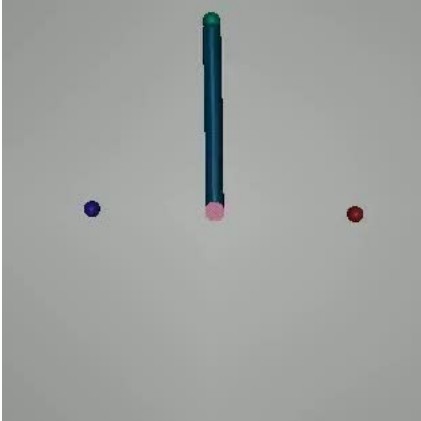

Figure 7: Our reaching toy environment with 2 target objects and **2 degrees of freedom (DOFs)**, with per-task randomized dynamics.

## B    REACHER ENVIRONMENT DETAILS

Figure 7 depicts our reaching toy environment. The agent controls a 2 degree of freedom (DOF) arm and should reach to one of the two randomly positioned objects. Crucially, the agent's dynamics are unknown and randomized at the start of each task: each of the two joints may have reversed orientation with 50% probability, and the agent's joint orientations may not match that of the task demonstration $\mathbf{d}_{i,k}$. This simulates a domain adaptive setting, for example the discrepancy in dynamics when a robot imitates from a human video demonstration. Since simply observing the demonstration is not sufficient to identify the task dynamics, the agent must use its own trials to identify the task dynamics and successfully reach the target object. Following (Brockman et al., 2016), the environment returns a reward at each timestep which penalizes the reacher's distance to the target object and the magnitude of its control inputs at that timestep. Precisely, let $\phi : \mathcal{S} \to \mathbb{R}^3$ map the agent's state to the reacher tip position, and $x_i \in \mathbb{R}^3$ be the true target object's position. Then:

$$r_i(s, a) = -||\phi(s) - x_i|| - ||a||^2 \qquad (5)$$

## C    EXPERIMENTAL DETAILS

We trained all policies using the ADAM optimizer (Kingma & Ba, 2015), on varying numbers of Nvidia Tesla P100 GPUs. Whenever there is more than 1 GPU, we use synchronized gradient updates and the batch size refers to the batch size of each individual GPU worker.

### C.1    REACHER ENVIRONMENT MODELS

For the reacher toy problem, every neural network in both the WTL policies and the baselines policies uses the same architecture: two hidden layers of 100 neurons each, with ReLU activations on each layer except the output layer, which has no activation.

Rather than mixture density networks, our BC and MIL policies are deterministic policies trained by standard mean squared error (MSE). The WTL Phase II policy $\pi^{\mathrm{II}}$ is also deterministic and trained by MSE, while the Phase I policy $\pi^{\mathrm{I}}$ policy is stochastic and samples actions from a Gaussian distribution with learned mean and diagonal covariance (equivalently, it is a simplification of the MDN to a single component). We found that a deterministic policy works best for maximizing the MIL baseline's average return, while a stochastic WTL trial policy achieves lower returns itself but facilitates easier learning in Phase II. Embedding architectures are also simplified: the demonstration "embedding" is simply the state of the last demonstration trajectory timestep, while the trial embedding architecture replaces the temporal convolution and flatten operations with a simple average over per-timestep trial embeddings.

We trained all policies for 50000 steps using a batch size of 100 tasks and a .001 learning rate, using a single GPU operating at 25 gradient steps per second.

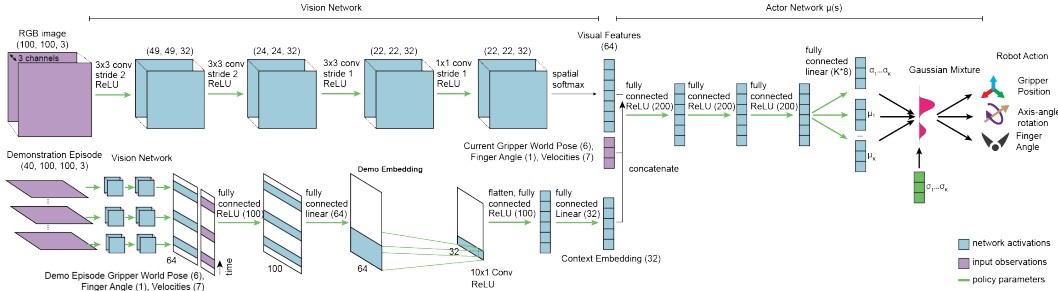

Figure 8: Our vision-based Phase I architecture. This is identical to the architecture in Figure 3, except that there are no trial embeddings or trial rewards to concatenate in the conditioning network.

Table 2: Best Model Hyperparameters for Gripper Environment Experiments

| METHOD | LEARNING RATE | MDN COMPONENTS |
|---|---|---|
| VISION MODELS | | |
| WTL | $9.693 \times 10^{-4}$ | 20 |
| MIL | $9.397 \times 10^{-4}$ | 20 |
| BC | $1.093 \times 10^{-3}$ | 20 |
| STATE-SPACE MODELS | | |
| WTL | $2.138 \times 10^{-3}$ | 20 |
| MIL | $2.659 \times 10^{-3}$ | 20 |
| BC | $2.138 \times 10^{-3}$ | 20 |

## C.2 GRIPPER ENVIRONMENT MODELS

---
**Algorithm 3** Demonstration and Trial Subsampling
---
1: **Input:** Demonstration $\mathbf{d} = \{(s_t, a_t)\}_{t=1}^T$ (or WLOG, trial $\tau$), for $T > 2$
2: **Input:** Output length $U = 40$
3: Initialize $\hat{\mathbf{d}} = \{(s_1, a_1), (s_T, a_T)\}$
4: **for** step $= 1, \cdots, U - 2$ **do**
5:      Sample $i$ uniformly at random from $\{2, \cdots, T - 1\}$, with replacement.
6:      $\hat{\mathbf{d}} \leftarrow \hat{\mathbf{d}} \cup \{(s_i, a_i)\}$
7: **end for**
8: **return** $\text{sort}(\hat{\mathbf{d}})$
---

The gripper environment policies illustrated in Figure 3 and Figure 8 produce context embeddings from demonstrations and (for Phase II) trial trajectories. The demonstrations and trials are variable length sequential data, since each gripper environment episode will terminate early if successful. To avoid dealing with variable sized inputs, we subsample 40 timesteps with replacement from any demonstration or trial trajectory to provide as input to our policies. We describe the subsampling process in Alg. 3. Since our gripper environment rewards are sparse and only nonzero on the final timestep, the subsampling process always selects the first and last timesteps. It selects the remaining 38 timesteps uniformly at random from the noninitial and nonterminal timesteps of the original demonstration or trial. It returns the 40 selected timesteps in sorted order for temporal consistency.

For each state-space and vision method we ran a simple hyperparameter sweep. We tried $K = 10$ and $K = 20$ MDN mixture components, and for each $K$ we sampled 3 learning rates log-uniformly from the range $[.001, .01]$ (state-space) or $[.0001, .01]$ (vision), leading to 6 hyperparameter experiments per method. We selected the best hyperparameters by average success rate on the meta-

validation tasks, see Table 2. We trained all gripper models with a 11 GPU workers and a batch size of 8 tasks for 500000 (state-space) and 60000 (vision) steps. Training the WTL trial and re-trial policies on the vision-based gripper environment takes 14 hours each.

### C.3 Algorithmic Complexity of WTL

WTL's trial and re-trial policies are trained off-policy via imitation learning, so only two phases of data collection are required: the set of demonstrations for training the trial policy, and the set of trial policy rollouts for training the re-trial policy. The time and sample complexity of data collection is linear in the number of tasks, for which only one demo and one trial is required per task. Furthermore, because the optimization of trial and re-trial policies are de-coupled into separate learning stages, the sample complexity is fixed with respect to hyperparameter sweeps. A fixed dataset of demos is used to obtain a good trial policy, and a fixed dataset of trials (from the trial policy) is used to obtain a good re-trial policy.

The forward-backward pass of the vision network is the dominant factor in the computational cost of training the vision-based Phase II architecture. Thus, the time complexity for a single SGD update to WTL is $O(T)$, where $T$ is the number of sub-sampled frames used to compute the demo and trial visual features for forming the contextual embedding. However, in practice the model size and number of sub-sampled frames $T = 40$ are small enough that computing embeddings for all frames in demos and trials can be vectorized efficiently on GPUs.

### C.4 BC + SAC Baseline Training Curve

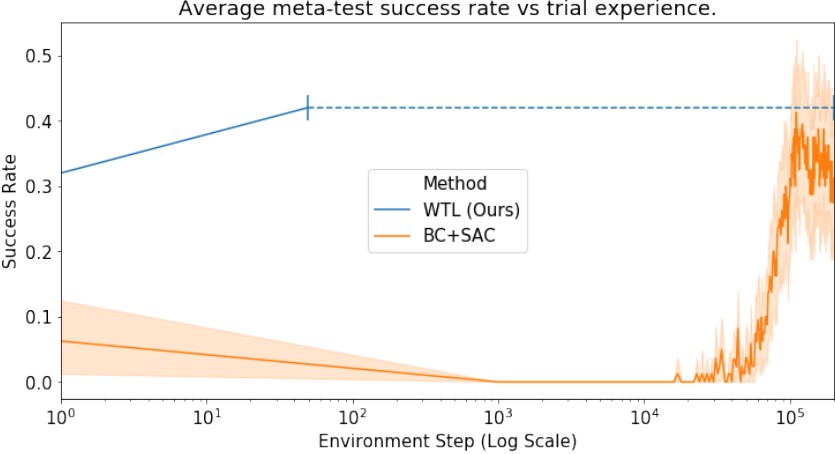

Figure 9: For "BC+SAC", we pre-train agents with behavior cloning and use RL to fine-tune on each meta-test task. By comparison, WTL uses a demo and a single trial episode per task ($<$ 50 environment steps). X-axis (log-scale) shows environment experience steps collected *per task*. Error bars indicate 95% confidence intervals.

## D Failure Analysis

We performed a qualitative failure analysis of our vision-based WTL method on meta-test tasks by manually inspecting videos of Phase I and Phase II policy behavior. Table 3 shows a matrix of different outcomes, where each episode is classified into one of three categories: 1) successfully completing the task, 2) performing the task but on the wrong object or location, 3) moves toward the correct object but misses (often comes close but gets stuck). One common failure mode is when the agent reaches towards but misses the correct object in Phase I, which does not provide the Phase II policy with enough information to disambiguate the correct task.

Table 3: WTL outcome matrix in the vision-based gripper environment. This table shows frequencies of outcomes after Phase I (rows) and after Phase II (columns) over 100 meta-test tasks.

|  |  | Phase II | | |
| --- | --- | --- | --- | --- |
|  |  | Missed Object | Wrong Object | Success |
| Phase I | Missed Object | 23 | 12 | 15 |
|  | Wrong Object | 5 | 11 | 8 |
|  | Success | 6 | 3 | 20 |

