# OpenReview forum: "Watch, Try, Learn: Meta-Learning from Demonstrations and Rewards"
_ICLR.cc/2020/Conference — Accept (Poster)_

### Official Review · AnonReviewer1 · 2019-10-20
**Official Blind Review #1**

**Rating:** 6

**Review:**

The paper proposes an approach for combining meta-imitation learning and learning from trial-and-error with sparse reward feedback. The paper is well-written and the experiments are convincing. I found the idea of having separate networks for the two phases of the algorithm (instead of recollecting on-policy trial trajectories) interesting and possibly applicable in other similar settings.

The paper has a comprehensive analysis of the advantages of the method over other reasonable baselines which do not have the trial-and-error element. However, in terms of the limitation of the method, the paper only has a small failure analysis section in the appendix; adding a more detailed discussion on the limitations can improve the impact of the paper further.

**Experience Assessment:**

I have read many papers in this area.

**Review Assessment: Checking Correctness Of Derivations And Theory:**

I did not assess the derivations or theory.

**Review Assessment: Checking Correctness Of Experiments:**

I assessed the sensibility of the experiments.

**Review Assessment: Thoroughness In Paper Reading:**

I made a quick assessment of this paper.

---

> ### Author Response · Authors · 2019-11-10
> **Updated with more discussion of limitation and future improvements**
>
> Thank you for the feedback. We address your concern below.
>
> >>> [...] adding a more detailed discussion on the limitations can improve the impact of the paper further.
>
> We’ve expanded the Discussion section to include a paragraph on limitations (as illustrated by our failure analysis) and future plans to address them.

---

### Official Review · AnonReviewer2 · 2019-10-23
**Official Blind Review #2**

**Rating:** 3

**Review:**

This work focuses on meta learning from both demonstrations and rewards. The proposed method has certain advantages over previous methods:
(1) In comparison to meta-imitation, it enables the agent to effectively and efficiently improve itself autonomously beyond the demonstration data.
(2) In comparison to meta-reinforcement learning, it can scale to substantially broader distributions of tasks, as the demonstration reduces the burden of exploration.

I am a bit confused by the writings and don't clearly understand the algorithm.
1. In eq. 4, should \theta be \phi?

2. What does \pi_\theta(a_t|s_t, {d_i,k}) mean? I'd like to see an example formulation of the policy. When the parameter \theta is well trained, does the policy still take demonstrations {d_i,k} as inputs? If yes, why are demonstrations needed?

3. In meta-testing (Algo. 2), will \theta and \phi be updated? Currently it looks like that the two parameters are fixed in meta testing. If so, this is a bit strange. Why not update the policies after sampling demonstrations (step 4) and collecting trials (step 5)?

4. "The state space S, reward ri, and dynamics Pi may vary across tasks." I think all the tasks should share the same input space; otherwise, the meta-trained two policies cannot be applied to a test task with different input space.  For example, if the states are 100x100 images in meta training, how to apply the policies to a meta-testing task with 100x200 images as states?


**Experience Assessment:**

I have published one or two papers in this area.

**Review Assessment: Checking Correctness Of Derivations And Theory:**

I carefully checked the derivations and theory.

**Review Assessment: Checking Correctness Of Experiments:**

I assessed the sensibility of the experiments.

**Review Assessment: Thoroughness In Paper Reading:**

I read the paper at least twice and used my best judgement in assessing the paper.

---

> ### Author Response · Authors · 2019-11-06
> **Updated submission to address questions**
>
> We thank the reviewer for the thoughtful comments and suggestions. We have included some corrections and re-uploaded the paper. To respond to the questions in order:
>
> 1. Thank you for catching this typo, \theta should be \phi in Eq. 4. We’ve corrected this in the updated paper.
>
> 2. We’ve updated Section 3.4 with a more concrete explanation of how the policies are implemented and to explain how the conditioning on demos or trials works. \pi_\theta(a_t|s_t, {d_i,k}) means that for any task, the phase I policy is conditioned on the task demonstrations (i.e., the policy infers the task from the demos). Concretely, the phase I policy has a demo embedding network which creates an embedding vector from the demo data, which we feed into the actor network that produces actions (bottom half of Fig. 3 and Fig. 7). Similarly, the phase II policy has both demo and trial embedding networks to condition on both demos and trial experience. Meta-training \theta and \phi amounts to meta-learning the weights of the embedding and actor networks for the phase I and II policies, respectively. The demonstrations are important even after meta-training \theta because at meta-test time the evaluation tasks are unknown prior to receiving the demos: the agent will be presented with a pair of new objects and the evaluation task could be to: a) pick one of them up, b) to slide one into the other, c) to pick and place, and so on. Conditioning on the demos helps the phase I policy infer what to do in the trial phase.
>
> 3. You are correct that the meta-parameters \theta and \phi are fixed during meta-testing. Since the policies infer the task using the embeddings, it is not necessary to update \theta or \phi within a task. We’ve updated Section 3.3 to clarify this.
>
> 4. You are correct that due to the architecture of the neural network policies, the state/observation dimension is fixed across tasks. However, the content of the state may vary between tasks: for example, the objects present in the image are different for each task. We’ve updated Section 3.4 to clarify this.

---

> ### Author Response · Authors · 2019-11-14
> **Please let us know if the response has addressed your concerns**
>
> Dear Reviewer 2,
>
> Could you let us know if our response has addressed the concerns raised in your review? We would be happy to provide further revisions or experiments to address any remaining issues, and would appreciate a response from you on the points that we raised.

---

### Official Review · AnonReviewer3 · 2019-10-26
**Official Blind Review #3**

**Rating:** 8

**Review:**

The paper introduces a meta-learning approach that can learn from demonstrations and subsequent reinforcement learning trials. The approach works in two phases. First, multiple demonstrations are collected for each task and a meta-learning policy is obtained by embedding the demonstrations of a single task in a context vector that is given to the actor policy.  Subsequently, the meta-policy is used to collect trajectories which are also evaluated with a (sparse) reward. These trajectories, along with the demonstrations, are again used by a second meta-learning policy (with a concatenated context from demonstrations and trajectories) to obtain the final policy. The algorithm is tested on a simple point reaching task as well as on a more complex 3d physics simulation that contains pick-and-place tasks, pushing tasks and button pressing.

The approach seems really interesting and I think combining demonstrations with reinforcement learning for meta learning is a very promising approach. This is also underlined by the experimental results presented in the paper. The only concern I have is the presentation of the paper. The algorithmic description is very short and many things are left unclear. I would also like to see a bit more ablation studies in the paper. More comments below:

- Just from the text it is not clear how the meta-learning actually works and you have to study the figures in detail to understand that a context is extracted from the demonstrations as well as the RL trajectories. The context should be better introduced (even though this has been shown already in different papers, I think it is a good strategy to write the
in a self-contained way).
- It is unclear to me how the reward signal is used to learn the meta-policy is phase 2. I understand that it is concatenated to the embedding before forming the context (again this is not really described in text but only in the figure), but it is quite unclear to me how the reward is used as optimality criteria. How do you reinforce good trajectories and decrease the influence of poor trajectories? This is maybe one of the most important parts of the paper and needs to be described in much more detail.
- The reward functions are not properly explained in the paper, even not for the toy task (reaching). This should at least be done in the appendix.
- I would like to see more ablation studies. For example, it is mentioned that 40 observations are used from the demonstrations and the trajectories. How is this number picked and how does it influence the performance? Also, typically, how long is a single trajectory? It is also unclear to me, if we take 40 random observations and the reward signal is sparse, wouldn't it be quite likely that the observations do not contain any reward values?
- How does K influence the performance of the algorithm?







**Experience Assessment:**

I have published one or two papers in this area.

**Review Assessment: Checking Correctness Of Derivations And Theory:**

I carefully checked the derivations and theory.

**Review Assessment: Checking Correctness Of Experiments:**

I carefully checked the experiments.

**Review Assessment: Thoroughness In Paper Reading:**

I read the paper thoroughly.

---

> ### Author Response · Authors · 2019-11-10
> **Submission updated to improve presentation**
>
> We thank the reviewer for the thoughtful comments, and have updated the paper to improve the presentation:
>
> >>> The context should be better introduced [...]
>
> We have updated Section 3.4 to explain in more detail how the policies condition on demonstration/trial data by extracting context vectors.
>
> >>> It is unclear to me how the reward signal is used to learn the meta-policy is phase 2 [...]
>
> As you noted, the phase II policy incorporates the trial reward information into the context embedding which is fed to the actor network that selects actions in phase II. It should learn how to use the reward information directly through the meta-training process. Suppose the trial trajectory has low reward, then the phase II policy should not reinforce the poor trial behavior otherwise it will receive a high loss for that task (Equation 3). We’ve updated the portion of Section 3.3 (right after Equation 3) to state this explicitly.
>
> >>> The reward functions are not properly explained in the paper [...]
>
> We’ve updated the Appendix A.3 and Appendix B to include our description of the environment cost functions.
>
> >>> If we take 40 random observations and the reward signal is sparse, wouldn't it be quite likely that the observations do not contain any reward values?
>
> We’ve updated Appendix A.3 & C.2 to clarify this. An episode has a maximum of 50 timesteps, corresponding to 5 seconds, but ends early if the task is successfully completed before then. As a result, most demonstrations are under 50 timesteps and we decided to sample 40 timesteps (with replacement) because most of the demonstrations, barring a few outliers, were <= 40 timesteps long. When sampling the 40 timesteps, we always included the last timestep at the end. Since our episodes terminate on success, the last timestep will contain the reward signal unless the agent failed.
> The practice of subsampling trajectories to be a fixed length follows prior works on one-shot imitation learning (e.g. Finn et al. One-Shot Visual Imitation Learning via Meta-Learning; James et al. Task Embedded Control Networks).

---

### Decision · Program_Chairs · 2019-12-19

**Decision:**

Accept (Poster)

**Comment:**

The paper proposed a meta-learning approach that learns from demonstrations and subsequent RL tasks.
The reviewers found this work interesting and promising. There have been some concerns regarding the clarity of presentation, which seems to be addressed in the revised version. Therefore, I recommend acceptance for this paper.